| Mean Sea Level and Tidal Change in Ireland since 1842: A case<br>study of Cork       Permatted: Left         David T. Pugh <sup>1</sup> , Edmund Bridge <sup>2</sup> , Robin Edwards <sup>3</sup> , Peter Hogarh <sup>1</sup> , Guy Westbrock <sup>4</sup> , Philip L.<br>Woodwork <sup>1</sup> , Gerard D. McCarthy <sup>2</sup> Permatted: Net Superscript Subscript <sup>5</sup> National Coencegrapty Centre, Joseph Provations Building, 6 Brownlow St, Liverpool, UK       Permatted: Net Superscript Subscript <sup>5</sup> National Coencegrapty Centry, Joseph Providem Building, 6 Brownlow St, Liverpool, UK       Permatted: Net Superscript Subscript <sup>6</sup> Office of Public Work, Jonathan Swin Breet, Trini, Co. Menh, Itelind       Permatted: Net Superscript Subscript <sup>10</sup> CARTS, Department of Geography, Maynooth, Co. Kildare, Ireland       Correspondence to: Gerard D. McCarthy (genuel incearthy@ims.ic)         Abstract. Knowledge of regional changes in mean sea level and local changes in tides are crucial to inform effective efimate adaptation. An essential element of the National Tide Gauge Network in the mid-2000s, is limited but belies a wealth or distortion data available in archival form. In this study, we digitize records located in CocK Harbow, Iredand from 1842 or minute change in the phase repectively, once adjustments for assensal and modal effects are made. Our results show tidal stability with a <sup>1</sup> / <sub>2</sub> change in the many Tork and Cock Harbow report and 4.         20       minute change in the phase vore protievel, yone cad distanter for sealable, we abow that with careful seasonal, nodal, and atmospheric corrections, together with knowledge of benchmark provenance, these historic, survey-oriented data can accurately inform of sea level changes.       Peleted: stocker et al. 2013                                                                                                                                                                                                                                                                                                                 |    |                                                                                                                                                                                                                                                                                                                                                                                                                                                                                                                                                           |   |
|-----------------------------------------------------------------------------------------------------------------------------------------------------------------------------------------------------------------------------------------------------------------------------------------------------------------------------------------------------------------------------------------------------------------------------------------------------------------------------------------------------------------------------------------------------------------------------------------------------------------------------------------------------------------------------------------------------------------------------------------------------------------------------------------------------------------------------------------------------------------------------------------------------------------------------------------------------------------------------------------------------------------------------------------------------------------------------------------------------------------------------------------------------------------------------------------------------------------------------------------------------------------------------------------------------------------------------------------------------------------------------------------------------------------------------------------------------------------------------------------------------------------------------------------------------------------------------------------------------------------------------------------------------------------------------------------------------------------------------------------------------------------------------------------------------------------------------------------------------------------------------------------------------------------------------------------------------------------------------------------------------------------------------------------------------------------------------------------------------------------------------------------------------------------------------------------------------------------------------------|----|-----------------------------------------------------------------------------------------------------------------------------------------------------------------------------------------------------------------------------------------------------------------------------------------------------------------------------------------------------------------------------------------------------------------------------------------------------------------------------------------------------------------------------------------------------------|---|
| study of Cork       David T, Pugh', Edmund Bridge <sup>2</sup> , Robin Edwards <sup>3</sup> , Peter Hogarth <sup>1</sup> <sub>2</sub> Guy Westbrock <sup>4</sup> , Philip L.       Formatted: Not Superscript Subscript         Study of Cork       Formatted: Not Superscript/Subscript       Formatted: Not Superscript/Subscript         Study of Cork       Formatted: Not Superscript/Subscript       Formatted: Not Superscript/Subscript         Study of Cork       Formatted: Not Superscript/Subscript       Formatted: Not Superscript/Subscript         Stool of Natural Sciences, Trinity College Dublin, Ireland       *The Marine Institute, Revirity. Maynooth, Co. Kildare, Ireland       Correspondence to: Gerard D. McCarthy (gerard necarthy@mx.is)         Abstract. Knowledge of regional changes in mean sea level and local changes in itides are crucial to inform effective climate adaptation. An essential element is the availability of accurate observations of sca level. Sca level data in the Republic of Ireland, prior to the establishment of the National Tide Gauge Network in the mid-2000s, is limited but belies a wealth of historical measurements is 1% and 2 minutes for amplitude and phase respectively, once adjustments for stansonal and nodal effects are made. Our mean sea level estimates are accurate to 2 on level, once adjustments for stansonal and nodal effects are made. Our mean sea level ternds plus local function and accurately inform tidal and wealth level and local, and amospheric corrections, together with knowledge of benchmark provenance, these historic, survey-oriented data can accurately inform of sea level function sea level (MSL), lide, and a non-tidal residual due to weather and other causes (Pagh and Woodworth 2014). Mean sea level (MSL), lide, and a non-tidal residual due to causal changes, and optentially due                                                                                                                                                                                                                                                                                                           |    | Mean Sea Level and Tidal Change in Ireland since 1842: A case                                                                                                                                                                                                                                                                                                                                                                                                                                                                                             |   |
| <ul> <li>Woodworth<sup>1</sup>/<sub>a</sub> Gerard D. McCarthy<sup>3</sup></li> <li>Formatted: Not Superscript/Subscript</li> <li>National Oceanography Centre, Joseph Proudman Building, 6 Brownlow St, Liverpool, UK <sup>5</sup>Office of Public Works, Jonathan Swift Steret, Trim, Co. Meath, Ireland <sup>15</sup>School of Natural Sciences, Trinity College Dublin, Ireland <sup>16</sup>The Marine Institute, Rivnine, Rivnike, Rivnine Callway, Ireland <sup>17</sup>CoRte: Of Public Works, Callway, Ireland <sup>17</sup>CoRte: Of Public Works, Callway, Ireland <sup>18</sup>CARUS, Department of Geography, Maynooth University, Maynooth, Co. Kildare, Ireland <sup>10</sup>Correspondence to: Gerard D. McCarthy (gerard-mecarthy/2mu.ic) </li> <li>Abstract. Knowledge of regional changes in mean sea level and local changes in tides are crucial to inform effective climate adaptation. An essential element is the availability of accurate observations of sea level. Sea level data in the Republic of Ireland, prior to the establishment of the National Tide Gauge Network in the mid-2000s, it limited but belies a wealch of historical data available in archival form. In this study, we digitize records located in Cork Harbour, Ireland from 1542 and 15 show how short duration (6-8 weeks), high quality data, with a large interval to the present, can accurately inform tidal and mean sea level estimates are accurate to 2 can level, once adjustments for seasonal and nodal effects are made. Our mean sea level estimates are accurate to 2 can level, once adjustments for atmospheric and seasonal, nodal, and an anotyperic corrections, together with knowledge of benchmark provenance, these historic, survey-oriented data can accurately inform of sea level changes.</li> <li>J Introduction</li> <li>25 Sea level is a combination of mean sea level (MSL), tide, and a non-tidal residual due to weather and other causes (Pugh and Woodworth 2014). Mean sea levels are rising globally as a consequence of anthropogenic climate change approximately inform of sea level changes.</li> <li>J Introduction</li> <li>25 Sea level is a combination of mean sea</li></ul> | 1  |                                                                                                                                                                                                                                                                                                                                                                                                                                                                                                                                                           |   |
| <ul> <li><sup>2</sup>Office of Public Works, Jonathan Swift Street, Trim, Co. Meath, Ireland</li> <li><sup>3</sup>School of Natural Sciences, Trimity College Dublin, Ireland</li> <li><sup>4</sup>TicARUS, Department of Geography, Maynooth University, Maynooth, Co. Kildare, Ireland</li> <li><sup>10</sup>Correspondence to: Geara D. McCarthy (general-incearthy/Ginu ie)</li> <li>Abstract. Knowledge of regional changes in mean sea level and local changes in tides are crucial to inform effective climate adaptation. An essential element is the availability of accurate observations of sea level. Sea level data in the Republic of Ireland, prior to the establishment of the National Tide Gauge Network in the mid-2000s, is limited but belies a wealth of historical data available in archival form. In this study, we digitize records located in Cork Harbour, Ireland from 1842 and book how shord duration (6-8 weeks), high quality data, with a large interval_to the present, can accurately inform tidal and nodal effects are made. Our results show tidal stability with a 2% change in the amplitude of the M2 component and 4-</li> <li><sup>10</sup>minute change in the phase over a period of 177 years; and mean sea level rise of 40 cm in the Cork Harbour area from 1842 (period the week show fad) stability with a 2% change in the amplitude of the M2 component and 4-</li> <li><sup>10</sup>minute change in the phase over a period of 177 years; and mean sea level rise of 40 cm in the Cork Harbour area from 1842 (period the since</li> <li><sup>10</sup>Deleted: since</li> <li><sup>10</sup>Deleted: since</li> <li><sup>10</sup>Deleted: since</li> <li><sup>10</sup>Sender Correspondence of the main sea level with knowledge of benchmark provenance, these historic, survey-oriented data can accurately inform of sea level changes.</li> <li><sup>11</sup>Introduction</li> <li><sup>12</sup>Sea level (Marg and area (Haigh et al. 2007), Jinicke et al. 2017). Knowledge of regional changes in mean sea level read construct of phanned</li> <li><sup>11</sup>Introduction</li> <li><sup>12</sup>Sea level (Ider et al. 2017). K</li></ul>                        |    |                                                                                                                                                                                                                                                                                                                                                                                                                                                                                                                                                           |   |
| Abstract. Knowledge of regional changes in mean sea level and local changes in tides are crucial to inform effective climate adaptation. An essential element is the availability of accurate observations of sea level. Sea level data in the Republic of Ireland, prior to the establishment of the National Tide Gauge Network in the mid-2000s, is limited but belies a wealth of historical data available in archival form. In this study, we digitize records located in Cork Harbour, Ireland from 1842 and 15 show how short duration (6–8 weeks), high quality data, with a large interval to the present, can accurately inform tidal and nedal effects are made. Our mean sea level estimates are accurate to 2 cm level, once adjustments for stanspheric and seasonal effects are made. Our mean sea level estimates are accurate to 2 cm level, once adjustments for stanspheric and seasonal effects are made. Our mean sea level trends plus local glacial isostatic adjustment. More broadly, we show that with careful seasonal, nodal, and atmospheric corrections, together with knowledge of benchmark provenance, these historic, survey-oriented data can accurately inform of sea level changes.       Deleted: since         25       Sea level is a combination of mean sea level (MSL), tide, and a non-tidal residual due to weather and other causes (Pugh and Woodworth 2014). Mean sea levels (MSL), tide, and a non-tidal residual due to weather and other causes (Pugh and Woodworth 2014). Mean sea levels for gional changes in mean sea level from global rends, and other singe allocal changes in mean sea level from global trends, and othanges in tides are crucial to inform effective local climate adaptation (Talke et al. 2013). An essential clement of planned                                                                                                                                                                                                                                                                                                                                                                                                                  | 5  | <sup>2</sup> Office of Public Works, Jonathan Swift Street, Trim, Co. Meath, Ireland<br><sup>3</sup> School of Natural Sciences, Trinity College Dublin, Ireland<br><sup>4</sup> The Marine Institute, Rinville, Oranmore, Galway, Ireland                                                                                                                                                                                                                                                                                                                |   |
| <ul> <li>adaptation. An essential element is the availability of accurate observations of sea level. Sea level data in the Republic of Ireland, prior to the establishment of the National Tide Gauge Network in the mid-2000s, is limited but belies a wealth of historical data available in archival form. In this study, we digitize records located in Cork Harbour, Ireland from 1842 and</li> <li>15 show how short duration (6-8 weeks), high quality data, with a large interval to the present, can accurately inform tidal and mean sea level changes. We consider error sources in detail. We estimate for the main M2 tidal constituent the accuracy of these historical measurements is 1% and 2 minutes for amplitude and phase respectively, once adjustments for seasonal and nodal effects are made. Our mean sea level estimates are accurate to 2 cm level, once adjustments for seasonal and nodal effects are made. Our nesults show tidal stability with a 2% change in the amplitude of the M2 component and 4-</li> <li>20 minute change in the phase over a period of 177 years; and mean sea level rends plus local glacial isostatic adjustment. More broadly, we show that with careful seasonal, nodal, and atmospheric corrections, together with knowledge of benchmark provenance, these historic, survey-oriented data can accurately inform of sea level changes.</li> <li>25 Sea level is a combination of mean sea level (MSL), tide, and a non-tidal residual due to weather and other causes (Pugh and Woodworth 2014). Mean sea level (MSL), tide, and a non-tidal residual due to weather and other causes (Pugh and Woodworth 2014). Mean sea levels are rising globally as a consequence of anthropogenic climate change <u>QPCC</u>, 2021) and tides are also changing in some local areas (Haigh et al. 2020; Jänicke et al. 2021), due to coastal changes, and potentially due to rising sea levels (Idier et al. 2017). Knowledge of regional changes in mean sea level from global trends, and changes in tides are crucia to inform effective local climate adaptation (Talke et a</li></ul>                                                                                     | 10 | 0 Correspondence to: Gerard D. McCarthy (gerard.mccarthy@mu.ie)                                                                                                                                                                                                                                                                                                                                                                                                                                                                                           |   |
| <ul> <li>these historical measurements is 1% and 2 minutes for amplitude and phase respectively, once adjustments for seasonal and nodal effects are made. Our mean sea level estimates are accurate to 2 cm level, once adjustments for atmospheric and seasonal effects are made. Our results show tidal stability with a 2% change in the amplitude of the M2 component and 4-</li> <li>20 minute change in the phase over a period of 177 years; and mean sea level rise of 40 cm in the Cork Harbour area from 1842 to 2019, approximately in line with global mean sea level trends plus local glacial isostatic adjustment. More broadly, we show that with careful seasonal, nodal, and atmospheric corrections, together with knowledge of benchmark provenance, these historic, survey-oriented data can accurately inform of sea level changes.</li> <li>25 Sea level is a combination of mean sea level (MSL), tide, and a non-tidal residual due to weather and other causes (Pugh and Woodworth 2014). Mean sea levels are rising globally as a consequence of anthropogenic climate change (PCC, 2021) and tides are also changing in some local areas (Haigh et al. 2020; Jänicke et al. 2021), due to coastal changes, and potentially due to rising sea levels (Idier et al. 2017). Knowledge of regional changes in mean sea level from global trends, and changes in tides are crucial to inform effective local climate adaptation (Talke et al. 2018). An essential element of planned</li> </ul>                                                                                                                                                                                                                                                                                                                                                                                                                                                                                                                                                                                                                                                                                                           | 15 | adaptation. An essential element is the availability of accurate observations of sea level. Sea level data in the Republic of<br>Ireland, prior to the establishment of the National Tide Gauge Network in the mid-2000s, is limited but belies a wealth of<br>historical data available in archival form. In this study, we digitize records located in Cork Harbour, Ireland from 1842 and<br>show how short duration (6–8 weeks), high quality data, with a large interval to the present, can accurately inform tidal and <b>Deleted:</b> (177 years) | ) |
| <ul> <li>to 2019, approximately in line with global mean sea level trends plus local glacial isostatic adjustment. More broadly, we show that with careful seasonal, nodal, and atmospheric corrections, together with knowledge of benchmark provenance, these historic, survey-oriented data can accurately inform of sea level changes.</li> <li>1 Introduction</li> <li>25 Sea level is a combination of mean sea level (MSL), tide, and a non-tidal residual due to weather and other causes (Pugh and Woodworth 2014). Mean sea levels are rising globally as a consequence of anthropogenic climate change (<u>IPCC, 2021</u>) and tides are also changing in some local areas (Haigh et al. 2020; Jänicke et al. 2021), due to coastal changes, and potentially due to rising sea levels (Idier et al. 2017). Knowledge of regional changes in mean sea level from global trends, and changes in tides are crucial to inform effective local climate adaptation (Talke et al. 2018). An essential element of planned</li> </ul>                                                                                                                                                                                                                                                                                                                                                                                                                                                                                                                                                                                                                                                                                                                                                                                                                                                                                                                                                                                                                                                                                                                                                                                           |    | these historical measurements is 1% and 2 minutes for amplitude and phase respectively, once adjustments for seasonal and nodal effects are made. Our mean sea level estimates are accurate to 2 cm level, once adjustments for atmospheric and                                                                                                                                                                                                                                                                                                           |   |
| 25 Sea level is a combination of mean sea level (MSL), tide, and a non-tidal residual due to weather and other causes (Pugh and<br>Woodworth 2014). Mean sea levels are rising globally as a consequence of anthropogenic climate change (PCC, 2021) and<br>tides are also changing in some local areas (Haigh et al. 2020; Jänicke et al. 2021), due to coastal changes, and potentially<br>due to rising sea levels (Idier et al. 2017). Knowledge of regional changes in mean sea level from global trends, and changes<br>in tides are crucial to inform effective local climate adaptation (Talke et al. 2018). An essential element of planned                                                                                                                                                                                                                                                                                                                                                                                                                                                                                                                                                                                                                                                                                                                                                                                                                                                                                                                                                                                                                                                                                                                                                                                                                                                                                                                                                                                                                                                                                                                                                                              | 20 | to 2019, approximately in line with global mean sea level trends plus local glacial isostatic adjustment. More broadly, we show that with careful seasonal, nodal, and atmospheric corrections, together with knowledge of benchmark provenance,                                                                                                                                                                                                                                                                                                          | ) |
| Woodworth 2014). Mean sea levels are rising globally as a consequence of anthropogenic climate change (PCC, 2021) and<br>tides are also changing in some local areas (Haigh et al. 2020; Jänicke et al. 2021), due to coastal changes, and potentially<br>due to rising sea levels (Idier et al. 2017). Knowledge of regional changes in mean sea level from global trends, and changes<br>in tides are crucial to inform effective local climate adaptation (Talke et al. 2018). An essential element of planned                                                                                                                                                                                                                                                                                                                                                                                                                                                                                                                                                                                                                                                                                                                                                                                                                                                                                                                                                                                                                                                                                                                                                                                                                                                                                                                                                                                                                                                                                                                                                                                                                                                                                                                 |    | 1 Introduction                                                                                                                                                                                                                                                                                                                                                                                                                                                                                                                                            |   |
| 30 adaptation is the availability of accurate observations of sea level.                                                                                                                                                                                                                                                                                                                                                                                                                                                                                                                                                                                                                                                                                                                                                                                                                                                                                                                                                                                                                                                                                                                                                                                                                                                                                                                                                                                                                                                                                                                                                                                                                                                                                                                                                                                                                                                                                                                                                                                                                                                                                                                                                          | 25 | Woodworth 2014). Mean sea levels are rising globally as a consequence of anthropogenic climate change ( <u>PCC, 2021</u> ) and<br>tides are also changing in some local areas (Haigh et al. 2020; Jänicke et al. 2021), due to coastal changes, and potentially<br>due to rising sea levels (Idier et al. 2017). Knowledge of regional changes in mean sea level from global trends, and changes                                                                                                                                                          |   |
|                                                                                                                                                                                                                                                                                                                                                                                                                                                                                                                                                                                                                                                                                                                                                                                                                                                                                                                                                                                                                                                                                                                                                                                                                                                                                                                                                                                                                                                                                                                                                                                                                                                                                                                                                                                                                                                                                                                                                                                                                                                                                                                                                                                                                                   | 30 | adaptation is the availability of accurate observations of sea level.                                                                                                                                                                                                                                                                                                                                                                                                                                                                                     |   |

- In the Republic of Ireland, a National Tide Gauge Network is maintained by the Office of Public Works and the Marine Institute with over 40 tidal gauges in total (Cámaro-García et al., 2021). This extensive network of sea level observations was established during the 2000s. Long-period mean sea level data prior to the establishment of the National Tide Gauge Network in the mid-2000s in the Republic of Ireland is limited to observations in Dublin (1938–2009) and Malin Head (1958–2002) (Holgate et al. 2013). No high frequency (e.g. hourly) tidal data are available prior to 2003. Most Irish sea level data are not openly accessible through global databases, meaning that Irish sea levels are not included in global studies of
- MSL, such as Frederikse et al. (2020) or of tides, such as Mawdsley et al. (2015) or Woodworth (2010).

Nevertheless, this lack of publicly available sea level data for Ireland belies a wealth of historical data that exists in various archival formats. Paper and digital images of marigraphs (traces recorded from automated tide gauges) exist for Dublin

- (1923–2003), Dún Laoghaire (formerly Kingstown, 1925–1933, S. Coate, Dún Laoghaire Harbourmaster, personal communication), Cork (from 1930, K. Hickey, UCC, personal communication) and Belfast (from 1901–2010, Murdy et al. 2015). The 1839-1843 National mapping and levelling survey of Ireland undertook a detailed sea level measurement exercise at 22 sites in the summer of 1842 (Figure 1, Airy 1845)—hereafter referred to as the Airy sites. These sites were connected by levelling during the survey (Cameron 1855), and potentially provide estimates of MSL at the 22 locations
  (Figure 1), Haughton (1856) made a nationwide survey of high and low waters for a period of a year in 1850–51 at twelve
- (Figure 1). Haughton (1856) made a nationwide survey of high and low waters for a period of a year in 1850–51 at twelve sites, six of which coincide with the Airy sites.

Although surveys such as those of Airy and Haughton are limited in time—8 weeks of continuous measurements in 1842, and upwards of one year of high and low waters in 1850–51, respectively—the long interval of over 150 years between these

- measurements and modern measurements indicates that long term MSL and tidal changes may be detected against natural year-on-year sea level variations. To this end we have initiated a full study of these historical data, especially the 22 Airy sites, and the 12 Haughton sites. This study will include making new measurements at many sites and requires a thorough assessment of MSL and tidal change based on these historical and recent data. As a first stage we have developed a multiinstitute pilot study of changes in Cork Harbour, as shown by comparing the Airy 1842 and our 2019 measurements.

Previous studies of long term MSL change in Ireland have generally focused on the north east of the country, from Dublin to Malin Head (the northernmost point in Ireland), due to the relative availability of data in this region. Relatively low rates of MSL rise (

Figure 1. Location of this study. Main map shows Cork Harbour, with the locations referred to in the text. (inset) Locations of all Airy sites on the island of Ireland are indicated with small black circles. Passage West, as the focus of this study, is highlighted with a red circle. R. Lee = River Lee. Data available from: Passage West (June-Aug 1842, June-July 2019), Roberts Cove (Jan-June 1973), Ballycotton (Oct 2010-present), Cobh (1906), Currach Club (Jun 2019-present), Ringaskiddy (Jan 2012-present).

In this paper, we focus our pilot study on Cork Harbour (Figure 1), with the joint aims of:

· developing a multi-institute approach to sea level studies in Ireland,

- proving the utility of historical data to inform long term change in sea level and tidal constituents,
- assessing evidence for changes in MSL, and tidal constituent changes, and
- · contextualising our results in terms of the impact on one of Ireland's most vulnerable coastal regions.

Formatted: Left

Measurements in 1842 were taken at Passage West 10 km south east of Cork City (Figure 1) as part of the Airy survey (Airy 1845). We have digitised these records and analysed them for tidal constituents and for MSL. These MSL results are then

compared with modern measurements from the nearby National Tide Gauge Network sites at Ringaskiddy, Cork Currach Club, and Ballycotton together with measurements at Cobh from the early 1900s, and observations at Robert's Cove in the 1970s. In order to determine the tidal constituent changes, we obtained a 6-week record of sea level at Airy's exact location in Passage West in June-July 2019.

#### 2 Case Study Location

- Cork City is located in the south of Ireland, where the River Lee meets Cork Harbour. Cork Harbour is one of the world's finest natural harbours. The Port of Cork has long been Ireland's largest southern port (Horgan 1955). Admiralty Charts from 1773 entitled Upper Harbour, and from 1777 entitled Lower Harbour and Approaches, show a channel from Cork City to the Celtic Sea, some 24 km in overall length, 11 km from Cork to Passage West and 13 km from Passage West to the open sea. The channel passes the current major deep-water port of Ringaskiddy, and then the former passenger port at Cobh and the
- present naval base at Haulbowline Island. Passage West was also formerly a passenger port and an outport for the city, prior to the deepening of the city quays commencing in 1894 (Rynne 2005). The channel depth upstream of Passage West is artificially maintained by dredging at 6.5 m below Chart Datum (CD), and to 5.2 metres upstream of the Tivoli Industrial Estate to Cork City. From Passage West towards the open ocean, the channel depth continues steady at about 15-20 metres depth, maintained by natural processes, including the 2 ms-1 tidal currents. Some limited land reclamation has occurred, for 105 example at the Chemical Works of Marino Point, and some harbour deepening to 11 metres CD at Ringaskiddy.

The tides in Cork Harbour are driven by the tides in the Celtic Sea, which travel from west to east, as a progressive Kelvin wave from the Atlantic Ocean towards the Irish Sea and Bristol Channel; the system is complicated by a weaker reflected Kelvin wave. Semidiurnal tides predominate, as elsewhere on the northwest European shelf, with diurnal tides very small.

- Around the entrance to Cork Harbour, the diurnal tides are only about two centimetres, as there is a local diurnal amphidrome. We will not refer to diurnal tides further in this study. As the semidiurnal tide spreads into Cork Harbour, there is a constraint on the flow due to the width and depth of the channels. The Admiralty Tide Tables (ATT) show time and height differences related to Cobh which is an ATT Standard Port for a large part of the southern Irish coast, for historical Admiralty charting reasons. The ATT indicates slightly later tides as the wave moves up the harbour, with small increases in 115 amplitude.

Cork City has a long history of flooding, by a combination of fluvial flooding from the River Lee and tidal flooding. Tidal flooding has become more common through the 20th century (Tyrrell and Hickey 1991). In 2009, the city suffered flooding losses of up to 100 million EUR (Hickey 2010). Cork City was part of an OPW Pilot Catchment Flood Risk Assessment and

- Management (CFRAM) study, that was substantially complete in November 2009, which identified the flood risk in Cork City and the basic structure of the preferred solution for the City. Substantive consultation and planning for flood alleviation resulted in the proposed Lower Lee Flood Relief Scheme. The scheme allows for improved management and control of the River Lee via upstream dams at Inniscarra and Carrigadrohid, provision of quayside defences, as well as flood defence embankments between Inniscarra and the City, and a system of pumps to remove additional water (Office of Public Works
- 2017a). The preferred solution was selected following consideration of a range of options, in both the Lower Lee Project and the CFRAM pilot project. The inflow from the ocean is uncontrolled in this plan. A Tidal Barrier was among the options considered in both the CFRAM and Lower Lee project, but it was not found to be feasible for a variety of reasons. Following proposal of the project, a Tidal barrier was again suggested during consultation by members of the public. Further consideration was given to the tidal barrier option, but it remained unfeasible, again for a variety of reasons. (Office of
- Public Works 2017b). Understanding the currently anticipated changes in future sea levels and tides, in the context of historical changes is important in preparing a project to defend Cork City against the current level of flood risk, as well as considering how adaptation can be built into such a project.

# **3 Data Collection**

Our primary purpose has been to compare 1842 and recent sea levels at Passage West. The 1842 levels were measured as 135 part of the 1839-1843 Ordnance Survey of Ireland (Cameron, 1855). The work included simultaneous sea level measurements at 22 Irish coastal locations from late June to late August 1842 (Figure 1). The results of Airy's analyses are published in Airy (1845) while the original observations are available in hand-written ledgers at the Cambridge University Library, Royal Greenwich Observatory archives (Figure 2). More details are in Andrews (1975, Appendix 3) and Dixon (1949). The 1842 survey consisted of readings every 5 minutes through the day, and over a few hours through the night-time 140 high water or low water.

Two observers, usually military people, were assigned to each site: at Passage West they were J Ludgate and Wm Dawson, on whose detailed work this study depends. A tide gauge (i.e., a graduated board) was nailed to the face of the steamboat wharf. Measurements relative to the gauge zero could in turn be related to the height of a copper bolt driven vertically into

one of the coping stones of the pier. This bolt could not be located in 2019 but the level of this bolt to Ordnance Datum 145 Dublin (ODD) from the 1839-1843 levelling is given in Cameron (1855) as 14.276 feet. We have digitised the hand-written ledgers, taking only values on the hour, which is adequate resolution for tidal work, and then made an analysis using software which can work with 19th century data.

Deleted: every