# Peer review of "(untitled)"

_Ocean Science, 2021_

## Author Response (AR1)

**Response to Review by Dr Martha Marcos**

*Responses in italics*

This manuscript succeeds in providing information on long term changes on tides and mean sea levels at Cork using only a short set of data recovered from historical archives in combination with modern sea level observations. Many details are provided on the data collection and on the corrections applied to make old and new data comparable. This is the core of the paper. Once this is achieved and the remaining uncertainties are estimated, the analyses of tidal changes and mean sea level trends are straightforward. Results are consistent with other works that point at only small (and local) changes in tides over the past 200 years and with earlier estimates of mean sea level trends in the region. I think this work deserves publication and I am looking forward to seeing the comprehensive study that the authors are planning to carry out at many more sites around Ireland.

*We would like to thank Dr Marcos for the review of our manuscript and encouragement for the wider project.*

I provide below a list of questions and suggestions, followed by some typos:

Sections 1, 2: I think it would be useful for the reader to summarise the information on stations and periods to facilitate the reading, together with Figure 1. It is easy to get lost in the text otherwise.

*We have added some annotation to the map caption: Passage West (June-Aug 1842, June–July 2019), Roberts Cove (Jan–June 1973), Ballycotton (Oct 2010–present), Cobh (1906), Currach Club (Jun 2019–present), Ringaskiddy (Jan 2012–present). This information is repeated in Table 3.*

Lines 134-135: how are 5-min readings converted into hourly? one value every hour has been kept and the other 5-min values disregarded, or have they been averaged?

*We have rewritten line 135 to clarify:" We have digitised the hand-written ledgers, **taking only values on the hour**, which is adequate resolution for tidal work, and then made an analysis using software which can work with 19$^{th}$ century data."*

Page 6, 1$^{st}$: if sea level is measured using a pressure gauge, then atmospheric pressure is probably also recorded. Air pressure observations are then mentioned in line 295.

*Atmospheric pressure is implicity recorded as part of the pressure data (there was no explicit second measurement of air pressure). None of these air pressure variations are significant at these latitudes at tidal frequencies, they do not affect the astronomical tidal analyses*

Section 4.1.1: Figure 4 shows, according to the caption, the seasonal cycle of the M2 modulation due to MA2 and MB2, but the text (line 190) refers to non-astronomical effects, which is contradictory.

*The reviewer comments about astronomical and non-astronomical parts of ~MA~2 and MB2 are correct and the text is muddled. We have removed "astronomic" in line 181, and "non-astronomical" in line 192.*

Line 216: estimation of the magnitude of the nodal modulation of M2 based on other sites. Are these listed somewhere?

*They are listed in line 205. In line 218 "…sites (Woodworth et al, 1991; Araujo 2005). We have repeated the references for clarity.*

Line 233: what are these uncertainties ? according to the text after them, seems to refer to interannual variability, but is it not specified.

*Yes, interannual variability considered here. We have clarified with "In assessing the tidal uncertainties, any tidal value measured over a short period may differ from the longer-term average because of real variations, such as natural interannual variability, and measurement errors. To consider interannual variability, we to look at Ringaskiddy 2012-2019".*

Line 295: see my comment above on the air pressures...

*Dealt with above.*

Lines 476-478: this seems a bit speculative since the 27 cm are an averaged value. It would make more sense to compare with closest stations (averaged or not)

*Unfortunately, no analysis of Irish stations exist over the relevant time period so we believe that the Hogarth et al. number from Britain is the best to compare with.*

Typos:

Line 147: is this reference to figure mistaken? Maybe figure 3...
Line 231: "we to look"
Lines 429-430: these two sentences are repetitive
Lines 473-475: please use the same number of digits, for consistency. Line 476: 40.2 cm

Reference Dwyer is incomplete
Reference Hogarth (2021) is already published Reference Horsburgh (2020) doi is missing

*We would like to thank Dr Marcos for these typos and reference issues, all of which have been corrected in the revised version.*

**Comment on os-2021-49**

Anonymous Referee #2

Referee comment on "Mean Sea Level and Tidal Change in Ireland since 1842: A case study of Cork" by David T. Pugh et al., Ocean Sci. Discuss., https://doi.org/10.5194/os-2021-49-RC2, 2021

Review of 'Mean Sea Level and Tidal Change in Ireland since 1842: A case study of Cork'

In this paper, the authors present the results from a tide-gauge data digitization effort in Cork, Ireland: data from a large field campaign in 1842 is digitized, and a levelling campaign has been undertaken to compare the historical measurements with present-day observations. The authors find a sea-level rise of about 40 cm over 177 years and a small but significant change in the amplitude and phase of the semi-diurnal tide.

I have enjoyed reading the paper, and the manuscript has taught me a lot on all the processes and uncertainties that are involved in tide-gauge data rescue efforts. I recommend publication in Ocean Sciences, and I'm convinced that the digitized records will have many use cases in the oceanic and geophysical community.

*We would like to thank the reviewer for taking the time to read and review the manuscript and for their generous comments.*

I have some (very) minor comments:

L41: The word 'Marigraphs', which I think refers to automatic tide gauges, might need a quick explainer.

*We will add the phrase* "(traces recorded from automated tide gauges)" *to the line in the revised manuscript.*

L420ff: an alternative to estimating the range of inter-annual variability could be to exploit the coherent interannual and decadal variability around the British Isles, as shown in Hogarth et al. 2021. The surveying period might have been during a period of below- average MSL values around the British Isles, or the other way round. Not sure how large this effect is though.

*The common mode developed by Hogarth et al. was developed on the basis of UK tide gauges. Their examination of satellite altimetry data certainly indicates that this is a common mode to both Britain and Ireland. However, the satellite data is only available from 1993 onwards. For our purposes, we don't think it is a robust index to consider the adjustments to Irish sea level data from 1842.*

*This is a very interesting mode of variability and it is our intention to analyse it in an Irish context but we believe the more conservative estimation of interannual variability that we have used here is more understandable in this context.*

L474: GIA uncertainty might be large in this region. For example, the GIA model from Caron et al.(2018) predicts a relative sea-level rise of 0.6 mm/yr for the region around Cork. This model is far from optimized for this region, as it's not using a sophisticated local deglaciation history, but the GIA signal might be a major reason for the difference between the rate from Hogarth et al. 2021 and the number found here.

*We have utilsed the most recent GIA model tailored for this region updated from Bradley et al. as part of the BritIceChrono project. This is a variation in the choice of GIA that was investigated in the PhD of Peter Hogarth. However, in terms of explaining the observed sea-level rise. Alternative GIA estimates such as the global estimates of Peltier give lower rates of relative sea level rise in the Cork region. Thus other models are less tuned for the region and explain less of the signal.*

*We envision that our follow up study that will cover the whole island of Ireland will allow a more detailed comparison of the various GIA models and their strengths and weaknesses.*

*We've added the revised text:* "Other GIA models such as the global model of Peltier and Tushingham (1991) results in an even lower contribution from GIA as would be expected for a model not tuned for the region."

Figures 1 and 6: R. Lee, does that refer to the river Lee?

*Yes, we will add (R. Lee = River Lee) to first caption.*

Finally, I'd encourage the authors to deposit the digitized time series and levelling information in a public repository, for example PSMSL or Zenodo.

*We welcome this suggestion. We will deposit the final data to Zenodo + add code to a github repository to recreate Figure 8. As a note, PSMSL are currently developing a format for submission of rescued data but this is not standardized at this stage.*